# Getting evidence into clinical practice: protocol for evaluation of the implementation of a home-based cardiac rehabilitation programme for patients with heart failure

Paulina Daw [1], Samantha B van Beurden [2,3], Colin Greaves [1], Jet J C S Veldhuijzen van Zanten [1], Alexander Harrison [4], Hasnain Dalal [3,5], Sinead T J McDonagh [3], Patrick J Doherty [4], Rod S Taylor [3,6]

PD and SBvB are joint first authors.

For numbered affiliations see end of article.

**Correspondence to**
Paulina Daw;
pxd891@student.bham.ac.uk

## ABSTRACT

**Introduction** Cardiac rehabilitation (CR) improves health-related quality of life and reduces hospital admissions. However, patients with heart failure (HF) often fail to attend centre-based CR programmes. Novel ways of delivering healthcare, such as home-based CR programmes, may improve uptake of CR. Rehabilitation EnAblement in CHronic Heart Failure (REACH-HF) is a new, effective and cost-effective home-based CR programme for people with HF. The aim of this prospective mixed-method implementation evaluation study is to assess the implementation of the REACH-HF CR programme in the UK National Health Service (NHS). The specific objectives are to (1) explore NHS staff perceptions of the barriers and facilitators to the implementation of REACH-HF, (2) assess the quality of delivery of the programme in real-life clinical settings, (3) consider the nature of any adaptation(s) made and how they might impact on intervention effectiveness and (4) compare real-world patient outcomes to those seen in a prior clinical trial.

**Methods and analysis** REACH-HF will be rolled out in four NHS CR centres across the UK. Three healthcare professionals from each site will be trained to deliver the 12-week programme. In-depth qualitative interviews and focus groups will be conducted with approximately 24 NHS professionals involved in delivering or commissioning the programme. Consultations for 48 patients (12 per site) will be audio recorded and scored using an intervention fidelity checklist. Outcomes routinely recorded in the National Audit of Cardiac Rehabilitation will be analysed and compared with outcomes from a recent randomised controlled trial: the Minnesota Living with HF Questionnaire and exercise capacity (Incremental Shuttle Walk Test). Qualitative research findings will be mapped onto the Normalisation Process Theory framework and presented in the form of a narrative synthesis. Results of the study will inform national roll-out of REACH-HF.

**Ethics and dissemination** The study (IRAS 261723) has received ethics approval from the South Central (Hampshire B) Research Ethics Committee (19/SC/0304). Written informed consent will be obtained from all health

## Strengths and limitations of this study

► This will be the first study to investigate the real-world implementation of a home-based cardiac rehabilitation programme in the UK and also to include the evaluation of the real-world clinical effectiveness of the programme.

► The study will use Normalisation Process Theory as a theoretical framework to guide data collection and interpretation.

► The qualitative findings will inform the development of an implementation manual for policymakers, planners, providers and commissioners of cardiac rehabilitation services for patients with heart failure.

► A possible limitation of the study is that the four centres that will be appointed to implement the REACH-HF programme are large, well-established cardiac rehabilitation treatment centres and might not be representative of the national cardiac rehabilitation landscape—a potential sample bias towards early adopters.

► This study may have limited generalisability outside the UK.

professionals and patients participating in the study. The research team will ensure that the study is conducted in accordance with the Declaration of Helsinki, the Data Protection Act 2018, General Data Protection Regulations and in accordance with the Research Governance Framework for Health and Social Care (2005). Findings will be published in scientific peer-reviewed journals and presented at local, national and international meetings to publicise and explain the research methods and findings to key audiences to facilitate the further uptake of the REACH-HF intervention.

## INTRODUCTION
### Heart failure
Approximately 900 000 people are affected by heart failure (HF) in the UK.[1] Due to an ageing population, HF is becoming a

national healthcare challenge.[2] HF has a high impact on both patients and society; it can reduce exercise tolerance and health-related quality of life (HRQoL), increase the risk of mortality and unplanned hospital admissions and is associated with high healthcare costs.[3] There is also a considerable burden on the friends and family of people with HF.[4] Exercise-based cardiac rehabilitation (CR) programmes have been shown to enhance HRQoL in patients with HF and reduce unplanned hospital admissions.[3 5] With sufficient adherence, these benefits are consistently achieved in trial settings with both centre-based and home-based CR.[3] Although the National Institute of Health and Care Excellence (NICE) recommends that all patients with HF receive CR,[6] due to the frailty and poor health of this clinical population, as well as dislike of group-based exercise and practical constraints (eg, transportation), participation in centre-based CR remains poor.[7] Underutilisation of CR among this clinical population has been highlighted in the 2010 NICE guideline, with the uptake of CR being much lower than predicted and estimated at 5.3%.[8]

## Rehabilitation EnAblement in CHronic Heart Failure

The Rehabilitation EnAblement in CHronic Heart Failure (REACH-HF) programme is a new CR programme for patients with HF and their caregivers, aimed at achieving better HRQoL in the comfort of the patient's home. The 12-week, facilitated, home-based intervention was code-veloped with patients, caregivers and clinicians,[9] using an intervention mapping approach.[10] In recent randomised controlled trials (RCTs), REACH-HF resulted in significant clinical improvements in HRQoL and was cost-effective, with a cost falling within the current National Health Service (NHS) tariff for CR in the UK.[11 12] REACH-HF therefore provides an affordable, evidence-based, patient-centred alternative to centre-based CR. This provides a way to address the latest NICE guidance recommendation that patients with HF are offered 'a personalised, exercise-based CR programme in a format and setting (at home, in the community or in the hospital) that is easily accessible for the person'.[6]

## Implementation science: negotiating the research-to-practice gap

Research and development within the NHS is world leading. However, the NHS falls short when scaling up well-evidenced innovations or good practice.[13] The spread of innovations and evidence-based interventions across the NHS and other healthcare systems is subjected to various challenges.[14] First, moving complex interventions from research settings to real-world clinical implementation is a slow process.[15] Some of the barriers slowing down this process include the characteristics of the intervention itself such as its usability or fit with the existing processes in the organisation. Beyond this, individual or organisational barriers include the attitudes towards change and the innovation itself, resources available, expertise, time and competing priorities.[16]

Second, following uptake, the same intervention does not always perform in exactly the same way across different organisations. For example, there may be differences in the characteristics of the people involved. In clinical trials, patients tend to be included based on predetermined criteria and such criteria are rigorously checked prior to study participation. However, in practice, a broader patient population may end up using the intervention. There may also be differences in the characteristics of the organisations delivering the intervention in terms of access to resources, staff and expertise, compared with those available in clinical trials. With these differences in population characteristics and access to resources, unplanned adaptations may occur to better fit the new context. This initially slows down the process of implementation and also means that the intervention is no longer delivered as it was under clinical trial conditions.[17] Such unplanned adaptations often result in the interventions initially failing to reproduce the results that are found within the context of RCTs.[18] With a varied and ever changing healthcare landscape, it is crucial to understand the full complexity of implementing innovations into real-world clinical practice.[19] It is particularly important to explore how much of the intervention can or cannot change (and in what ways) without jeopardising the benefits of the intervention.[20]

Healthcare evaluations and improvement projects often consider performance at the level of individual healthcare professional,[21] targeting the professional's knowledge, routines and attitudes.[22] However, there is a need for wider reaching system-level evaluations of the implementation process that also take into account community, organisational, system-level and policy-level influences.[23]

Overall, implementation science aims to examine the process of implementation of healthcare innovations, in particular, the barriers and facilitators, as observed in real-life clinical settings.[24] To narrow the research-to-practice gap, implementation scientists recommend that the process of implementation is considered and built into the intervention design and development, the context and systems of implementation are assessed during the implementation efforts and key stakeholders are involved in the intervention development stage through to dissemination, implementation and evaluation.[23]

## Aims of the project

The current project aims to implement REACH-HF in four UK NHS CR services to (1) explore the facilitators of, and barriers to, implementation of REACH-HF in the existing UK CR services, (2) assess the implementation fidelity, (3) the extent and nature of any potential adaptations to the intervention content and how such adaptations impact on effectiveness and (4) compare real-world outcomes to the clinical trial findings.

## METHODS AND ANALYSIS

### Design

We will conduct a mixed-method implementation evaluation study using in-depth semistructured interviews with key NHS staff, analysis of pre–post intervention changes in routinely collected outcome data via the British Heart Foundation founded National Audit of Cardiac Rehabilitation (NACR) and a fidelity assessment using a checklist applied to recordings of provider–patient interactions.

In-depth semistructured interviews will be used to identify facilitators of, and barriers to, implementation. Audio recordings of REACH-HF clinical encounters will be used to assess fidelity. Quantitative data obtained from the NACR will be used to compare real-world outcomes to the clinical trial findings. Data gathered from all of the above study activities (interviews, fidelity assessment, patient outcomes) will be used to assess the extent and nature of adaptations to the intervention content and how such adaptations are associated with effectiveness.

### Setting and site recruitment

The study will be conducted in four UK NHS CR centres (desirably from the four UK countries) which will be early adopters of the REACH-HF programme and known as 'Beacon Sites'. The opportunity to apply to become a Beacon Site will be promoted at national (UK) conferences and local meetings of CR practitioners. Interested CR services will be sent an information pack including an application form. Applicants will be asked to provide information on their NACR National Certification Programme for CR status (NCP_CR), number of referrals made to the CR service (for both cardiac patients and patients with a primary diagnosis of HF), whether the service is offering home-based programme, length of current programmes, number of programme completions, number of pre and post-treatment assessment completions, as well as to comment on willingness to engage in research and host site visits for other interested parties.

The NCP_CR is a national certification programme for CR issued jointly by the British Association for Cardiovascular Prevention and Rehabilitation (BACPR) and the NACR. The certification programme rates CR services on seven key performance indicators (KPIs). KPIs are the NACR measurable indicators based on the BACPR core components. Programmes need to meet at least four KPIs to be granted an amber status and all seven to be granted a green status (2019 NACR Quality and Outcomes report).

The sites will be recruited from across the UK using a two-stage application process (application form followed by panel interview for shortlisted sites). As an incentive, sites will be offered free intervention materials for the treatment of 50 patients (ie, the REACH-HF patient manual, the Family and Friends Resource, audio with relaxation techniques and chair-based exercise digital versatile disc (DVD)). In addition, the selected sites will be offered free training (including training manuals) for three health professionals to deliver REACH-HF, post-training support and formative feedback on

performance. The 3-day training will be delivered by the Heart Manual Department (HMD), NHS Lothian in Edinburgh.

To be eligible, sites have to be:

► NACR electronically registered sites with high-quality status from the past audit period (green or amber status) operating in the UK.
► Committed to delivering REACH-HF to 50 patients over the 12-month Beacon Site project period.
► Able to release three healthcare professionals (or more) with relevant experience in CR and/or HF for 3 days training plus one self-directed pretraining day.
► Able to engage in research to evaluate performance (ie, recording some intervention sessions and staff participation in interviews).
► Willing to host site visits and/or share information and/or experiences with other interested NHS parties.
► Conduct baseline and post-treatment assessment of HRQoL using the Minnesota Living with Heart Failure Questionnaire (MLHFQ)[25] and exercise capacity using the Incremental Shuttle Walk Test (ISWT)[26] for all patients receiving the REACH-HF programme.

### Study population

Healthcare providers: we aim to recruit up to 24 healthcare professionals. The total number will include the 12 health professionals delivering REACH-HF and other key NHS staff involved in the delivery, planning and commissioning of CR for patients with HF. To identify key staff involved in CR services, the study will use a combination of opportunity sampling (all available staff trained to deliver the REACH-HF programme) and snowball sampling (staff who are identified by the existing participants as having a key role in delivering or commissioning of CR).[27] This sampling strategy will be applied until saturation in the themes and concepts generated in the qualitative analysis is reached.

Patients: the study will include up to 200 patients with HF who are referred to the CR centres for rehabilitation and receive REACH-HF treatment. Out of the 200 patients, CR consultations of up to 48 patients (12 per site) receiving REACH-HF intervention will be audio recorded.

### Intervention

REACH-HF is a home-based, health professional facilitated, 12-week CR programme supporting self-care in patients with HF, which has been codeveloped with patients, caregivers and clinicians. The programme is described in detail elsewhere[11 12 28–30] and is summarised below.

The programme consists of:

► The Heart Failure Manual for the patient provides information about HF to increase understanding of the condition and address common misconceptions, information about and strategies for managing the condition, and further information related to HF,

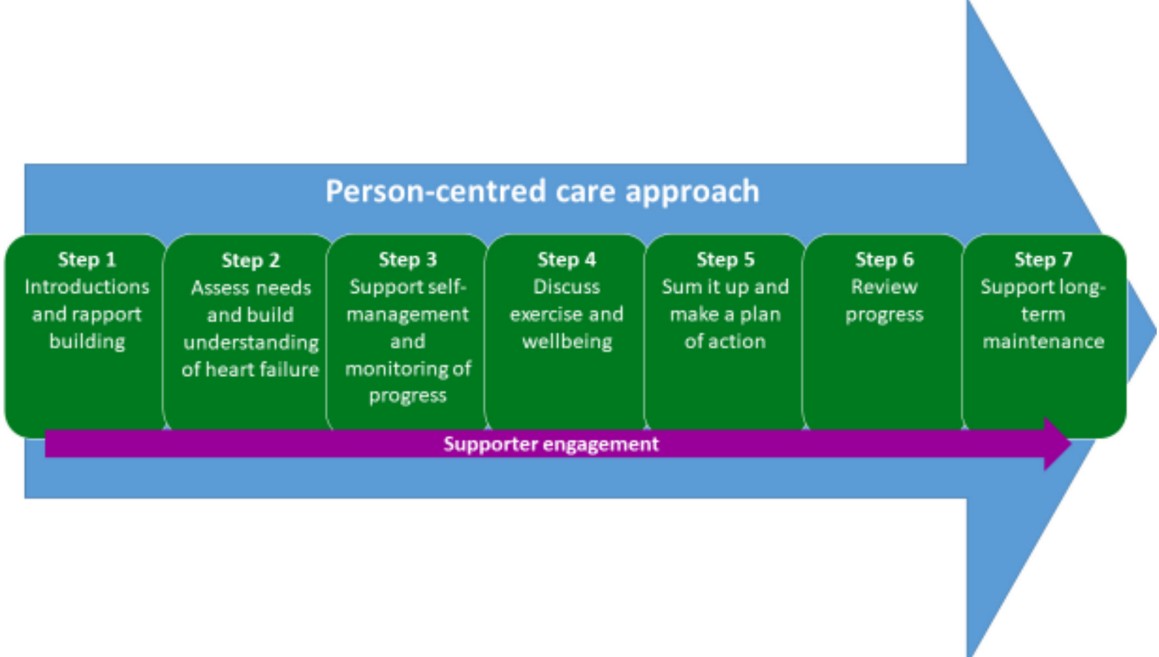

**Figure 1** The seven steps of successful REACH-HF facilitation. REACH-HF, Rehabilitation EnAblement in CHronic Heart Failure.

such as lifestyle risk management, managing depression and anxiety and getting support from others.

► A choice of two exercise training programmes; a chair-based programme (available on DVD and online) and a walking programme. Patients are recommended to engage in exercise three times per week, in addition to general physical activity.

► A stress management programme, with relaxation techniques, provided in the manual and in audio format, to help cope with anxiety and depression.

► A progress tracker designed for the patient to facilitate learning from experience through self-monitoring of behaviour and symptoms—prompting help-seeking, where necessary.

► A family and friends resource to increase caregiver understanding of the condition, to enable them to support the patient in their self-care and to help them address their well-being.

► Face-to-face and telephone facilitation over 12 weeks by a health professional trained to deliver the REACH-HF programme.

### Facilitator training

Three health professionals with CR and/or HF experience from each Beacon Site will attend a 3-day training course delivered by the HMD in Edinburgh. This training course will focus on the seven steps of successful facilitation of REACH-HF (figure 1) and include sessions on psychology, behaviour change, physical activity and exercise, engaging the caregiver and further content/interaction designed to bring all of the components together.

The Beacon Sites will determine which members of the CR team will attend the REACH-HF training. The

main requirement for the healthcare professional is the experience of delivering CR and/or of working with patients with HF. The facilitators will likely be HF/cardiac specialist nurses or physiotherapists/exercise specialists with qualifications and/or experience in the delivery of exercise-based CR programmes.

It is expected that site identification, training and set-up will take approximately 6 months. Following the set-up period, the Beacon Sites will have 12 months to deliver REACH-HF to 50 patients, during that time, qualitative interviews and audio recordings of REACH-HF sessions for selected patients will take place. At the end of Beacon Site activity, a quantitative data download will be requested from the NACR and an interim download will be requested 9 months from the end of the study to allow piloting of data-cleaning and processing procedures (stopping short of analysis).

### Measures and procedures
#### Qualitative interviews

In-depth semistructured interviews and focus groups with NHS staff to include REACH-HF practitioners (physiotherapists and CR nurses with experience in delivering centre-based CR, who had been trained to deliver the REACH-HF programme in a 3-day training course), service managers, clinical leads and commissioners. Interviews will take place at each Beacon Site (see online supplementary appendix 1 for the topic guide). Each identified staff member will, if possible, be interviewed twice (once at the beginning and once at the end of the data collection window) and one focus group will be held in each locality with identified study participants (at the midpoint of the data collection window). Interviews will

**Table 1** Qualitative questions and their origins in the NPT construct and components

| NPT construct | Construct's components | Interview questions |
|---|---|---|
| Coherence (sense-making) | Differentiation | Can you describe the REACH-HF intervention and how it differs from your usual way of working? |
| | Communal specification | What is your colleagues understanding of the purpose of the REACH-HF intervention? |
| | Individual specification | How does the intervention affect the nature of your work? |
| | Internalisation | In your opinion, what it the value of the REACH-HF intervention? To you? To your patients? |
| Cognitive participation (relational work) | Initiation | Who are the individuals (you can include yourself) that drive REACH-HF forward and get others involved? What are their roles? What are they doing to support the project? |
| | Enrolment | How did the team need to change in order to introduce REACH-HF? |
| | Legitimation | How do you feel about being involved in the REACH-HF project? |
| | Activation | What is the future of REACH-HF in your service? What factors can enable the integration of REACH-HF into a cardiac rehabilitation service? |
| Collective action (operational work) | Interactional workability | How easy or difficult has it been to integrate REACH-HF into your existing work? |
| | Relational integration | How has implementing REACH-HF affected working relationships within the team? |
| | Skills and workability | How do the skills of the staff delivering REACH-HF match the needs of the programme? |
| | Contextual integration | Was REACH-HF training sufficient to allow for successful implementation? If not, what other topics or skills could have been included? Are there enough resources available to support the REACH-HF programme? Are there any other barriers to delivering REACH-HF on your patch? |
| Reflexive monitoring (appraisal work) | Systematisation | Are you in any way evaluating effectiveness, usefulness or impact of REACH-HF on the service? |
| | Communal appraisal | Do your colleagues consider the intervention worthwhile? |
| | Individual appraisal | Do you consider it worthwhile? |
| | Reconfiguration | Can the REACH-HF intervention be easily modified and improved to suit your way of working? If yes, in what way? |

NPT, Normalisation Process Theory; REACH-HF, Rehabilitation EnAblement in CHronic Heart Failure.

be either face-to-face or by phone. The development of topic guides for qualitative interviews and focus groups was based on 4 constructs and 16 subdomains from the Normalisation Process Theory (NPT) framework (table 1). The topic guides content may be amended depending on feedback from stakeholders and the first few interviews.

Two video-conferencing peer supervision sessions will be available to all REACH-HF trained facilitators, provided by the HMD, as part of the REACH-HF training package. The researchers will observe and take notes from each of these sessions.

### Fidelity assessment

All REACH-HF CR treatment sessions (four–six contacts), both face-to-face and phone-based, of approximately 48 consenting patients (12 per site), will be audio recorded by the healthcare professionals delivering the programme. Each REACH-HF facilitator will be requested to audio record all treatment sessions for four patients with HF. The selection of which patients to include will be guided by the researchers, using a quasi-random process. Five months after the REACH-HF training, facilitators will be asked to invite all subsequent patients to take part in the study, until two willing patients with HF agree to have their treatment sessions recorded. Approximately 10 months after the REACH-HF training, an email will be sent to repeat the invitation and audio recording process for the next two consenting patients.

The quality of delivery (intervention fidelity) of the recorded treatments will be assessed by the researcher (PD) using the same fidelity checklist used in the original REACH-HF research study.[11] This will allow comparison with fidelity scores achieved in the clinical trial. The recordings for the first six patients will also be double scored and two researchers (PD and CG) will discuss any differences in their scores to agree and 'anchor' the

scoring process and minimise coder bias. If an agreement cannot be reached, a third reviewer (JJCSVvZ) will be appointed for arbitration.

The fidelity checklist is a 12-item checklist focused on identifying key delivery processes such as the use of a patient-centred communication style, making a plan of action and encouraging self-monitoring of progress (particularly with the exercise programme). The checklist uses the Dreyfus scale of clinical skill acquisition,[31] to rate clinical skills on a scale of 0–6 and is anchored such that a score of 3 or more represents adequate delivery quality for each item. Coding instructions are provided (online supplementary appendix 2).

REACH-HF facilitators will be asked to complete a brief self-rated fidelity checklist after each session they have recorded. This comprises questions about the same 12 main components of the treatment and allows the facilitators to rate the occurrences of each feature (absence, minimal, some, sufficient, good, very good, excellent) (online supplementary appendix 3). The main reason for including a self-rated fidelity checklist is that an independent observer rating is time-consuming/labour intensive, whereas a self-rating assessment might provide a pragmatic, lower cost alternative for checking delivery quality for use in real-world clinical practice.

Finally, for each patient opting into the study, age, sex, time since diagnosis and severity of symptoms will be recorded by the healthcare professionals delivering the REACH-HF intervention.

### Quantitative

At the end of the Beacon Site project period, a report will be requested from the NACR team based on the University of York on:

► Number of referrals made to the Beacon Sites during the study period.
► Number of patients with HF enrolled on the REACH-HF programme (attending at least one session).
► CR attendance (average number of face-to-face and telephone sessions per patient).
► Number of patients completing the REACH-HF programme (in the clinical trial[11] patient adherence was defined as attendance at the first face-to-face contact with the facilitator and at least two facilitator contacts thereafter—at least one of which must have been face-to-face).

Summary data on key pre and post-programme measures will also be requested to enable comparison with changes in the intervention group observed in the clinical trial. These include HRQoL—determined using the MLHFQ and exercise capacity—determined using the ISWT. The MLHFQ consists of 21 questions that rate on a scale of 0–5 (where 0 is not at all, 1 is very little and 5 is very much) how different HF symptoms (ie, swelling of ankles and legs, shortness of breath or tiredness, fatigue and poor energy levels) prevent the patient from living as they would have wanted to during the 4-week period prior

to the first CR session. ISWT is an externally paced exercise capacity test that can be administered in the field with minimal equipment and without medical supervision. The test has good test–retest reliability and it is an acceptable alternative to (widely used to assess physical fitness and functional capacity of cardiac patients) exercise test with ECG monitoring or the cardiopulmonary exercise test.[32] A recent study confirmed that a single ISWT is a valid, low resource, assessment of an estimate for physical fitness and functional capacity for CR patients.[33]

### Data analysis

#### Qualitative data

Digital recordings of interviews and focus groups will be transcribed verbatim and any potentially identifiable information, such as individual or location names, will be redacted. The transcripts (Word documents) will be uploaded into NVivo software to help organise the data for analysis.[34] Illustrative quotes, that may be used in future presentations or publications, will be presented alongside pseudonyms to protect anonymity.

The transcripts will be analysed according to the principles of framework analysis outlined by Ritchie and Spencer[35] and using the four over-arching constructs of NPT (coherence, cognitive participation, collective action and reflexive monitoring) as an initial framework for coding the data.[36] NPT suggests general mechanisms that are associated with successful implementation. These include service providers' understanding of the new intervention and how it differs from standard practice, their motivation and attitude towards the healthcare innovation and the work they do to deliver and evaluate the intervention. NPT will provide a framework for generating questions for interviews and focus groups and analysing gathered data. See table 1 for more details on the application of NPT to the data collection.

#### Fidelity assessment

Implementation fidelity scores from the fidelity checklist will be collated at the level of the facilitator, the site and the total sample and presented using descriptive statistics (means, ranges) using the same analytic approach as the original REACH-HF trial.[11] Numerical data (0–6) from the Dreyfus scale of clinical skill acquisition will be converted into categorical (yes/no) data reflecting whether the session reached the adequate level of delivery (score 3 or above). Observer-rated treatment fidelity will be compared with self-rated fidelity from the post-session fidelity questionnaires completed by the REACH-HF facilitators at the end of each recorded session. The analytic approach to compare the two rating scales will be Pearson's correlation for continuous scores[37] and Gwet's first-order agreement coefficient (the AC1 statistic) for categorical ratings.[38]

The fidelity assessment data sample reflects the sample size used to assess fidelity in the original REACH-HF clinical trial. We require a minimum of four patient

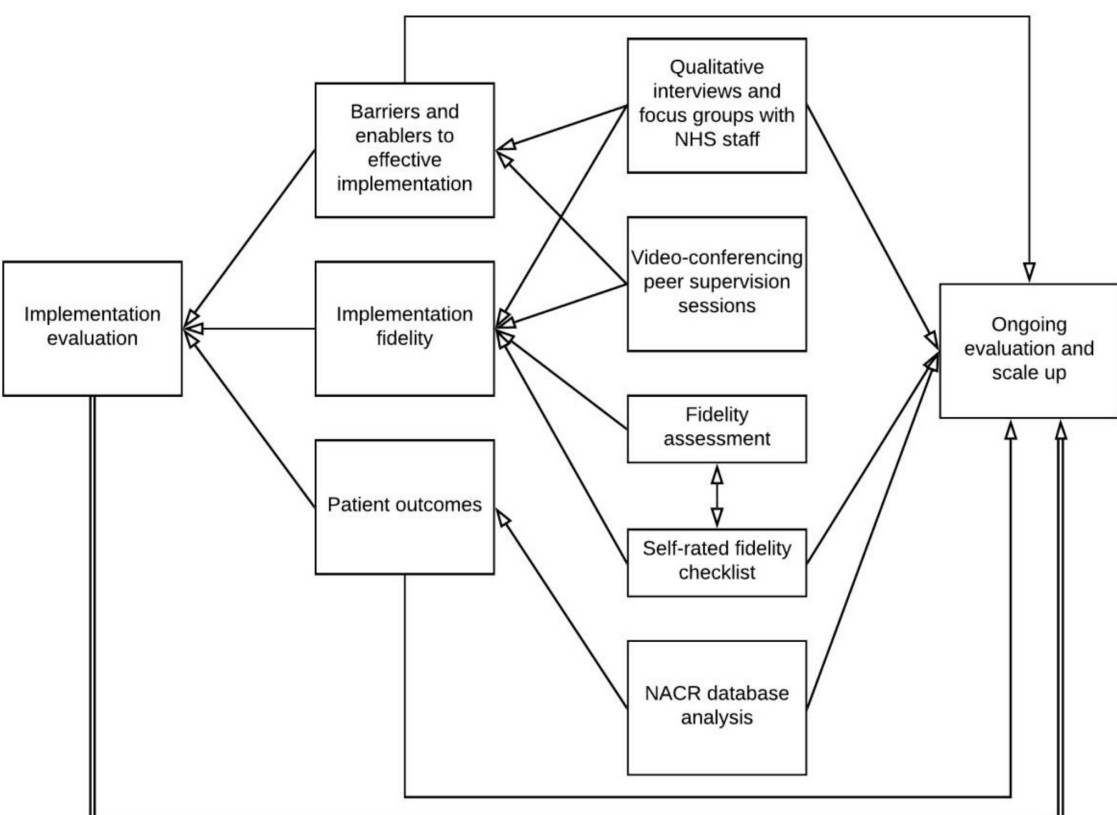

**Figure 2** Beacon Site evaluation and embedded processes for ongoing monitoring. NACR, National Audit of Cardiac Rehabilitation; NHS, National Health Service.

recordings per facilitator to be able to assess variation in performance between staff and between NHS sites.

### Quantitative outcomes

Changes from pre to post-treatment in outcome data (MLHFQ and ISWT) will be reported as mean scores with 95% CI within each Beacon Site. Mean change scores for patients receiving REACH-HF will be compared across Beacon Sites and also with the changes found in the REACH-HF trial. This comparison will take account of potential differences on patient characteristic and take due attention to the confidence intervals. Similarly, change scores for patients receiving REACH-HF will be compared with an aggregate change score from the NACR database for those who receive other forms of CR (primarily centre-based or digital CR). Subgroup analyses will be conducted by the NACR team to determine variations in uptake and outcomes within our REACH-HF cohort by site, sex and other characteristics of interest (eg, area deprivation index, rurality). Data on the number of patients treated, uptake and completion rates and session attendance, will be presented using descriptive statistics. Figure 2 illustrates the interactions between the study's aims and methods and how they link with the process of ongoing evaluation and scale-up.

### Patient and public involvement

Patient preference and acceptability have been addressed extensively during the REACH-HF clinical trials.[11 12] Six patients with HF and four caregivers have been consulted and informed the design of the REACH-HF programme. Patient and public involvement in the proposed study has included involving a member of the public to read and comment on the content of the study invitation letter, participant information sheet and the consent form designed for the study. Additionally, members of all CR teams involved in the study were consulted during the process of setting up the Beacon Sites on issues such as the feasibility of the study, selected outcome measures and the burden of participation in the study. At the end of the study, the final report will be shared with NHS staff at the participating Beacon Sites, allowing them to use it for service evaluation, future service planning and sharing of good practice.

### DISCUSSION

The research-to-practice translation gap is well documented. It is common that evidence-based interventions are not adopted into clinical settings and do not become routine practice. To narrow the translation gap, more insight is needed into mechanisms that allow for successful implementation of effective and cost-effective interventions. To advance the field, implementation theories and mechanisms need to be tested in real-world clinical settings.

The REACH-HF Beacon Site project is a multifaceted and interactive approach to a phased roll-out that aims to disseminate the multicentre trial findings, increase awareness of the REACH-HF intervention and explore replicability of the intervention in new contexts. At the time of writing this protocol, four Beacon Sites in Scotland have been established and will contribute further data on the implementation of REACH-HF.[39]

In line with earlier recommendations for implementation research, this study will open a channel of feedback between researchers and implementers (NHS staff), with a common goal of improved service delivery for patients with HF. This study will provide an insight into the translation of the REACH-HF clinical trial findings into real-world practice and an in-depth understanding of the implementation process in the context of current NHS provision. These findings will inform the future, larger-scale implementation of REACH-HF, offer guidance to policymakers, planners and commissioners of CR services, inform adaptations to the REACH-HF training package and intervention and facilitate adoption and spread of home-based CR for patients with HF in the UK.

**Author affiliations**
¹School of Sport, Exercise & Rehabilitation Sciences, University of Birmingham, Birmingham, UK
²Psychology, University of Exeter, Exeter, UK
³College of Medicine and Health, University of Exeter, Exeter, UK
⁴Health Sciences, University of York, York, UK
⁵Royal Cornwall Hospitals NHS Trust, Cornwall, UK
⁶MRC/CSO Social and Public Health Sciences Unit & Robertson Centre for Biostatistics, Institute of Health and Well Being, University of Glasgow, Glasgow, UK

**Correction notice** This article has been corrected since it was published. The incorrect trial registarion number has been removed.

**Contributors** All authors contributed to the idea for the study. PD and SBvB drafted the manuscript. SBvB led the setup and recruitment of Beacon Sites. STJMcD is overseeing the day-to-day management of the Beacon Site project. PD secured all relevant ethical approvals for the project and prepared all study documentation. CG, JJCSVvZ, HD and RST are providing project supervision and oversight. PJD and AH will coordinate access to the NACR data. AH provided statistical analysis advice. PD will acquire and analyse the data for the study. All authors provided critical revision of the manuscript for important intellectual content and approved the final draft of the protocol for submission.

**Funding** This study/project was funded by the National Institute for Health Research (NIHR) (Programme Grants for Applied Research scheme (project reference RP-PG-1210-12004)). The views expressed are those of the authors and not necessarily those of the NIHR or the Department of Health and Social Care. PD's time is funded by a PhD studentship from the University of Birmingham.

**Competing interests** None declared.

**Patient and public involvement** Patients and/or the public were involved in the design, or conduct, or reporting, or dissemination plans of this research. Refer to the Methods section for further details.

**Patient consent for publication** Not required.

**Provenance and peer review** Not commissioned; externally peer reviewed.

**ORCID iDs**
Paulina Daw http://orcid.org/0000-0002-0942-3953
Samantha B van Beurden http://orcid.org/0000-0001-7848-2159
Colin Greaves http://orcid.org/0000-0003-4425-2691
Jet J C S Veldhuijzen van Zanten http://orcid.org/0000-0001-8422-9512
Alexander Harrison http://orcid.org/0000-0002-2257-6508
Hasnain Dalal http://orcid.org/0000-0002-7316-7544
Sinead T J McDonagh http://orcid.org/0000-0002-0283-3095
Patrick J Doherty http://orcid.org/0000-0002-1887-0237
Rod S Taylor http://orcid.org/0000-0002-3043-6011

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
