## [Reviewer comments · BMJ Open]

ARTICLE DETAILS

TITLE (PROVISIONAL)	Getting evidence into clinical practice: Protocol for evaluation of the implementation of a home-based cardiac rehabilitation programme for patients with heart failure.
AUTHORS	Daw, Paulina; van Beurden, Samantha; Greaves, Colin; Veldhuijzen van Zanten, Jet; Harrison, Alexander; Dalal, Hasnain; McDonagh, Sinead; Doherty, Patrick Joseph; Taylor, Rod

VERSION 1 – REVIEW

REVIEWER	WEN-CHIH WU Brown University, Providence, RI, USA
REVIEW RETURNED	25-Jan-2020

GENERAL COMMENTS	This is an overall well designed implementation study. I have a few comments: 1. The study has four well established aims, yet the analytic plan is not explicitly matched to the individual aims and leaves the reader speculating which goes with what.2. There is no description of the statistical tests or methodology that will be utilized to compare the proposed quantitative outcomes and whether adjustment for covariates and accounting for clustering are necessary and why.3. There is no clear delineation of the study timelines from site identification, training and facilitation, to post-intervention assessment
---

REVIEWER	Dr. Kirstine L. Sibilitz University Hospital Copenhagen, Rigshospitalet Department of Cardiology, Denmark
REVIEW RETURNED	24-Mar-2020

GENERAL COMMENTS	The authors present a protocol for evaluating implementation of a recently published cardiac rehabilitation programme for patients with heart failure based on data from a RCT, in order to narrow the research-to-practice translation gap. The protocol is well written, with clearly defined and described methods and outcomes. The authors should be thanked for this very inspiring, well planned and innovative study, which is highly lacking in cardiac rehabilitation research. I have some minor comments: 1) In finding the Beacon Sites - how do the authors ensure to include a broad spectrum of sites and not only the most
--

	enthusiastic sites. There is a risk of selection bias. Please describe. 2) What does the coloring status for the sites mean? Amber and green status. Please explain. 3) The healthcare professionals - what groups/professions are included? Please specify. 4) How did the authors reach the number of 200 patients to be included? 5) Could the authors please elaborate a bit on the stress management programme. 6) The qualitative interviews: will the respondents be interviewed by the same interviewer the two times or two different persons? Please describe possible biases/limitations in the interviewing technique. 7) The quantitative part: what is a minimum of sessions for the intervention to have succeeded in the implementation? You have included for some outcomes but not all, please fulfill in all aspects. 8) What are the general limitations and biases of the implementation study. 9) Small revision, spelling; in the NPT construct section: cognitive part: Activation future of....(not or)
--	---

VERSION 1 – AUTHOR RESPONSE

We thank the reviewers for these very positive and encouraging comments.

Reviewer: 1

Reviewer Name: WEN-CHIH WU

Institution and Country: Brown University, Providence, RI, USA Please state any competing interests or state ‘None declared’: None

This is an overall well designed implementation study. I have a few comments:

1. The study has four well established aims, yet the analytic plan is not explicitly matched to the individual aims and leaves the reader speculating which goes with what.

We thank the reviewer for the comment and have added the following to the manuscript (“Design” section, line 231):

“In-depth semi-structured interviews will be used to identify facilitators of, and barriers to, implementation, audio-recordings of REACH-HF clinical encounters will be used to assess fidelity. Quantitative data obtained from the NACR will be used to compare real-world outcomes to the clinical trial findings. Data gathered from all of the study activities (interviews, fidelity assessment, patient outcomes) will be used to assess the extent and nature of adaptations to the intervention content and how such adaptations are associated with effectiveness.”

2. There is no description of the statistical tests or methodology that will be utilized to compare the proposed quantitative outcomes and whether adjustment for covariates and accounting for clustering are necessary and why.

We thank the reviewer for the comment and have added the following to the manuscript ("Fidelity assessment" section, line 435):

"Changes from pre- to post-treatment in outcome data (MLHFQ and ISWT) will be reported as mean scores with 95 % confidence intervals within each Beacon Site. Mean change scores for patients receiving REACH-HF will be compared across Beacon Sites and also with the changes found in the REACH-HF trial. This comparison will take account of potential differences on patient characteristic and take due attention to the confidence intervals. Similarly, change scores for patients receiving REACH-HF will be compared with an aggregate change score from the NACR database for those who receive other forms of CR (primarily centre-based or digital CR). Sub-group analyses will be conducted by the NACR team to determine variations in uptake and outcomes within our REACH-HF cohort by site, sex, and other characteristics of interest (e.g. area deprivation index, rurality). Data on the number of patients treated, uptake and completion rates and session attendance, will be presented using descriptive statistics."

Due to the nature of the available data, we do not plan to conduct any further statistical tests.

3. There is no clear delineation of the study timelines from site identification, training and facilitation, to post-intervention assessment

We thank the reviewer for the comment and have added the following to the manuscript ("Facilitator training" section, line 325):

"It is expected that site identification, training and set up will take approximately six months. Following the set up period, the Beacon Sites will have 12 months to deliver REACH-HF to 50 patients, during that time qualitative interviews and audio-recordings of REACH-HF sessions for selected patients will take place. At the end of Beacon Site activity, a quantitative data download will be requested from the NACR and an interim download will be requested 9 months from the end of the study to allow piloting of data-cleaning and processing procedures (stopping short of analysis)."

Reviewer: 2

Reviewer Name: Dr. Kirstine L. Sibilitz

Institution and Country: University Hospital Copenhagen, Rigshospitalet, Department of Cardiology, Denmark.

Please state any competing interests or state 'None declared': None.

The authors present a protocol for evaluating implementation of a recently published cardiac rehabilitation programme for patients with heart failure based on data from a RCT, in order to narrow the research-to-practice translation gap.

The protocol is well written, with clearly defined and described methods and outcomes. The authors should be thanked for this very inspiring, well planned and innovative study, which is highly lacking in cardiac rehabilitation research.

I have some minor comments:

1) In finding the Beacon Sites - how do the authors ensure to include a broad spectrum of sites and not only the most enthusiastic sites. There is a risk of selection bias. Please describe.

We thank the reviewer for the comment.

Our two main selection criteria were the site's ability to deliver REACH-HF within the 12-month time frame (access to HF patient) and quality of the NACR data. Also we sought to have a geographic spread of sites – ideally one in each of the 4 UK countries.

We added to the bullet point in the “Strengths and limitations of this study” section (line 134):

“ – a potential sample bias towards early adopters.”

We added to the manuscript (“Setting and Site Recruitment” section, line 238):

“desirably from the four UK countries”

2) What does the coloring status for the sites mean? Amber and green status. Please explain.

We thank the reviewers for the comment and have added the following information to the manuscript (“Setting and Site Recruitment” section, line 247):

“The NCP_CR is a national certification programme for CR issued jointly by the British Association for Cardiovascular Prevention and Rehabilitation (BACPR) and the NACR. The certification programme rates cardiac rehabilitation services on seven Key Performance Indicators (KPIs). KPIs are the NACR measurable indicators based on the BACPR core components. Programmes need to meet at least four KPIs to be granted an amber status and all seven to be granted a green status (2019 NACR Quality and Outcomes report).”

3) The healthcare professionals - what groups/professions are included? Please specify.

We thank the reviewer for the comment and have added the following information to the manuscript (“Qualitative interviews” section, line 333):

“NHS staff will include REACH-HF practitioners (physiotherapists and cardiac rehabilitation nurses with experience in delivering centre-based CR, who had been trained to deliver the REACH-HF programme in a 3-day training course), service managers, clinical leads and commissioners.”

The same healthcare professionals were included in the REACH-HF clinical trial.

4) How did the authors reach the number of 200 patients to be included?

We thank the reviewer for the comment.

The final sample target is comparable with the number of patients (216) randomised in the HFrEF trial, so we could replicate the primary outcome to the results of the RCT, as well as based on estimated throughput of patients based on around 2/3 of the treatment rate achieved in the REACH-HF trial (to ensure feasibility). Our additional constraint regarding patient recruitment was the availability of funds for staff training and patient materials (REACH-HF manuals, DVDs, CDs).

5) Could the authors please elaborate a bit on the stress management programme.

We thank the reviewer for the comment.

The stress management programme consists of three main elements:

1. Assessment of stress levels (via self-administered quiz within the manual) and indirectly via use of the HADS or other screening tools for anxiety and depression that are used in usual CR practice.
2. Discussion of ideas about ways to reduce /prevent stress in day to day life (text and ideas in the manual and through discussion with the facilitator)
3. Supporting the choice and use of a stress-management technique with options including simple breathing techniques, progressive physical relaxation and mindful breathing (thought using audio provided on CDs).

Ideas for self-delivery of cognitive behavioural techniques are included for helping patients to manage low mood or anxiety. However, the facilitators are trained to identify possible clinical levels of anxiety or depression and support a referral for specialist treatment and support where these are identified.

Intervention development paper can be accessed here:

<https://pilotfeasibilitystudies.biomedcentral.com/articles/10.1186/s40814-016-0075-x>

6) The qualitative interviews: will the respondents be interviewed by the same interviewer the two times or two different persons? Please describe possible biases/limitations in the interviewing technique.

We thank the reviewer for the comment.

Study participants will be interviewed by the same interviewer twice. That can potentially lead to an over familiarisation with participants/interview questions/data. On the other hand, using the same interviewer twice can enable the interviewer to build a rapport with the study participants facilitating the interviewing process.

7) The quantitative part: what is a minimum of sessions for the intervention to have succeeded in the implementation? You have included for some outcomes but not all, please fulfill in all aspects.

We thank the reviewer for the comment and have added the following information to the manuscript ("Quantitative" section, line 387):

"in the clinical trial patient adherence was defined as attendance at the first face-to-face contact with the facilitator and at least two facilitator contacts thereafter – at least one of which must have been face to face."

8) What are the general limitations and biases of the implementation study.

We thank the reviewer for the comment and have stated in the manuscript ("Strength and limitation of this study" section, line 122):

- "This will be the first study to investigate the real-world implementation of a home-based cardiac rehabilitation programme in the UK and also include the evaluation of the real-world clinical effectiveness of the programme.
- The study will use Normalisation Process Theory as a theoretical framework to guide data collection and interpretation.
- The qualitative findings will inform the development of an implementation manual for policy-makers, planners, providers and commissioners of cardiac rehabilitation services for heart failure patients.
- A possible limitation of the study is that the four centres that will be appointed to implement REACH-HF are large, well-established cardiac rehabilitation treatment centres and might not be representative of the national cardiac rehabilitation landscape – a potential sample bias towards early adopters.

- This study may have limited generalisability outside the UK.”

A non-randomised comparison could be listed as an additional limitation. However, the intention of the study is to audit adherence to good practice rather than to (re)establish effectiveness. Therefore this point is excluded from the list above.

9) Small revision, spelling; in the NPT construct section: cognitive part: Activation future of...(not or)

We thank the reviewer for the comment – we have corrected the typo.

VERSION 2 – REVIEW

REVIEWER	Wu, Wen-Chih Providence VA Medical Center, The Miriam Hospital Brown University
REVIEW RETURNED	11-May-2020

GENERAL COMMENTS	Minor grammar mistake on lines 468-469 of the paper: "At the time of writing this protocol, a further four Beacon sites in Scotland have been established and will also being contributing data..."
---

REVIEWER	Kirstine Lærum Sibilitz Department of Cardiology University Hospital Copenhagen Blegdamsvej 9 2100 Copenhagen
REVIEW RETURNED	05-May-2020

GENERAL COMMENTS	Thank you for a very well written manuscript. I have no further comments and my requests are fully addressed.
---